# M2N: Mesh Movement Networks for PDE Solvers

**Wenbin Song** [*]
ShanghaiTech University
`songwb@shanghaitech.edu.cn`

**Mingrui Zhang** [*]
Imperial College London
`mingrui.zhang18@imperial.ac.uk`

**Joseph G. Wallwork**
Imperial College London
`j.wallwork16@imperial.ac.uk`

**Junpeng Gao**
ETH Zürich
`jungao@student.ethz.ch`

**Zheng Tian**
ShanghaiTech University
`zheng.tian.11@ucl.ac.uk`

**Fanglei Sun**
ShanghaiTech University
`sunfl@shanghaitech.edu.cn`

**Matthew D. Piggott**
Imperial College London
`m.d.piggott@imperial.ac.uk`

**Junqing Chen**
Tsinghua University
`jqchen@tsinghua.edu.cn`

**Zuoqiang Shi**
Tsinghua University
`zqshi@tsinghua.edu.cn`

**Xiang Chen** [†]
Noah's Ark Lab, Huawei
`xiangchen.ai@outlook.com`

**Jun Wang** [†]
University College London
`jun.wang@cs.ucl.ac.uk`

## Abstract

Numerical Partial Differential Equation (PDE) solvers often require discretizing the physical domain by using a mesh. Mesh movement methods provide the capability to improve the accuracy of the numerical solution without introducing extra computational burden to the PDE solver, by increasing mesh resolution where the solution is not well-resolved, whilst reducing unnecessary resolution elsewhere. However, sophisticated mesh movement methods, such as the Monge-Ampère method, generally require the solution of auxiliary equations. These solutions can be extremely expensive to compute when the mesh needs to be adapted frequently. In this paper, we propose to the best of our knowledge the first learning-based end-to-end mesh movement framework for PDE solvers. Key requirements of learning-based mesh movement methods are: alleviating mesh tangling, boundary consistency, and generalization to mesh with different resolutions. To achieve these goals, we introduce the neural spline model and the graph attention network (GAT) into our models respectively. While the Neural-Spline based model provides more flexibility for large mesh deformation, the GAT based model can handle domains with more complicated shapes and is better at performing delicate local deformation. We validate our methods on stationary and time-dependent, linear and non-linear equations, as well as regularly and irregularly shaped domains. Compared to the traditional Monge-Ampère method, our approach can greatly accelerate the mesh adaptation process by three to four orders of magnitude, whilst achieving comparable numerical error reduction.

---

[*]Equal contribution.
[†]Corresponding Authors: Xiang Chen and Jun Wang.

36th Conference on Neural Information Processing Systems (NeurIPS 2022).

# 1 Introduction

Partial Differential Equations (PDEs) are widely used to model natural phenomena, ranging from astrophysics and ocean dynamics to semiconductor device simulation and bio-engineering [Evans, 2010]. Acquiring accurate numerical solutions efficiently for complex PDEs is an essential but challenging problem in all scientific and engineering disciplines. Solving PDEs using numerical methods such as the Finite Element Method (FEM) requires discretizing the problem spatially and temporally. A mesh is often used for spatial discretization and its quality affects the accuracy of the numerical solution [Frey and George, 2007]. However, it is often prohibitively expensive to solve the problem on a very high resolution mesh. Mesh adaptation is an advanced discretization method designed to tackle this problem. It increases the mesh resolution where the solution requires higher numerical accuracy, while decreasing the mesh resolution where unnecessary. Mesh adaptation methods can be generally divided into two categories: $h$-adaptation and $r$-adaptation. In $h$-adaptation, new mesh nodes are dynamically added to the regions where fine resolution is required. In $r$-adaptation (or *mesh movement*), however, mesh nodes are only relocated or moved without changing the mesh topology [Huang and Russell, 2011]. Compared to $h$-adaptation, $r$-adaptation has several attractive features. First, no extra mesh points are generated, which keeps the dimension of the linear system representing the discretized PDE unchanged. In addition, fixed mesh connectivity can also make the structure of the stiffness matrix unchanged, which enables matrix pre-factorization to accelerate the solution of the large linear systems encountered in FEM [Budd et al., 2009]. However, a common problem of mesh movement methods is mesh tangling, in which lines connecting the mesh nodes come across each other. Mesh movement methods based on optimal transport theory can effectively prevent mesh tangling issues (see [Clare et al., 2022] for a discussion on this), but require solving a Monge-Ampère equation at each adaptation step, which is highly time-inefficient.

AI-powered approaches to computing solutions of PDEs have been an emerging topic in recent years, and show great potential in solving problems where traditional numerical PDE solvers struggle (e.g. high dimensional problems [Han et al., 2018, Sheng and Yang, 2021]), or in accelerating the solution process by learning a neural operator from the parameterized description of a PDE problem to its corresponding solution [Li et al., 2020, Lu et al., 2019]. However, these methods still encounter fatal bottlenecks, such as the precision guarantee of the solution, data efficiency, generalization capability, etc. These are fundamental limits of deep learning, but essential for scientific computing scenarios.

A possible alternative is to perform learning-based mesh adaptation, by which the traditional numerical PDE solvers can achieve better performance, while the time consumption of mesh adaptation is greatly reduced. There have been some works in this direction [Zhang et al., 2020, Yang et al., 2021, Huang et al., 2021, Fidkowski and Chen, 2021, Tingfan et al., 2021, Pfaff et al., 2020]. However, most previous methods focus on mesh generation [Zhang et al., 2020] or mesh refinement [Yang et al., 2021, Huang et al., 2021, Fidkowski and Chen, 2021], instead of topology-invariant mesh movement as considered in this work. Moreover, the prior works are not end-to-end approaches, which means the neural networks are used to predict certain metrics, such as the local mesh density [Zhang et al., 2020, Huang et al., 2021] or the metric tensor [Fidkowski and Chen, 2021, Tingfan et al., 2021, Pfaff et al., 2020], which then have to be fed into a traditional mesher/remesher to obtain the mesh. Therefore, the overall performance is bounded by the mesher/remesher, which is computational geometry based and hence not optimized for solving PDE problems. On the contrary, in our method, the adapted mesh is directly output by the neural network.

In this work, we propose to the best of our knowledge the first learning-based end-to-end mesh movement framework for PDE solvers. Taking the source term, the input field, and/or the PDE parameters as input features, the model deforms an initial mesh to the adapted mesh by mesh movement. In usage, the model can be applied to a class of PDEs without retraining. We design a Neural-Spline based model for mesh deformation. It is an invertible neural network and hence can avoid mesh tangling. Moreover, its mechanism naturally guarantees that a hypercubic boundary can be maintained through learnable mapping. We also design a graph attention network (GAT) based model for mesh deformation. The graph neural network can naturally describe domains with irregular shapes and embed the relevant information. We utilize the attention mechanism of the GAT model to guarantee each mesh node stays within its neighborhood so that mesh tangling can be alleviated.

Our main contributions are listed as follows:

1. We propose a learning-based end-to-end mesh movement framework for PDE solvers, which to the best of our knowledge is the first of its kind. Without interfering with the PDE solver, the models can achieve numerical error reduction similar to the traditional Monge-Ampère method, while the mesh generation is accelerated by three to four orders of magnitude.

2. A Neural-Spline based model and a GAT based model are proposed for mesh deformation. Besides generalization to different PDE parameters, source terms, input solution fields, etc., the models are designed to guarantee boundary consistency, alleviate mesh tangling, and generalize to different mesh densities, which are all desired for mesh movement.

## 2   Related Work

**Mesh movement method.**   Mesh movement methods include velocity-based and location-based methods. In the work of [Anderson and Rai, 1983, Gnoffo, 1982, Farhat et al., 1998], the mesh is moved according to attraction and repulsion pseudo-forces between nodes motivated by a spring model. The moving mesh finite element method [Baines et al., 2005] computes the solution and the mesh simultaneously by minimizing the residual of the PDEs written in a finite element form. As for location-based method, the moving mesh PDE (MMPDE) method [Huang and Russell, 2011] moves the mesh through the gradient flow equation of an adaptation functional. In recent years, there has been a growing interest in optimally-transported $r$-adapted meshes [Budd and Williams., 2009, Clare et al., 2022].

**AI for PDE.**   To solve a PDE problem, neural networks can be used to represent the function to solve, and trained either with the residual loss of the PDE or using the variational principle [Raissi et al., 2019, Yu et al., 2017]. Neural operators are proposed to learn an operator from the problem function to the solution function [Li et al., 2020, Lu et al., 2019]. There also exist mesh-based PDE solvers with deep learning. In [Pfaff et al., 2020], a graph neural network with additional world edges is applied to predict dynamical systems, shown to be effective with a wide range of physical systems. In [Belbute-Peres et al., 2020], a differentiable PDE solver is embedded in a neural network to help predict accurate solutions and also backpropagate the loss so that the input coarse mesh can be optimized.

**AI for Meshing.**   AI methods have also been proposed for mesh generation, adaptation, and so on. MeshingNet [Zhang et al., 2020] uses a neural network to learn the required local mesh density, which can then be provided to a Delaunay triangulation based mesh generator to generate high-quality meshes. The optimal local mesh density is also learned in [Huang et al., 2021] for mesh refinement. On the other hand, the mesh refinement process is formulated as a reinforcement learning problem in [Yang et al., 2021] to minimize the PDE solution error under given refinement budgets. The flow field is predicted by machine learning models to calculate the metric tensor so that the mesh can be optimized accordingly [Tingfan et al., 2021]. The authors in [Fidkowski and Chen, 2021] focused on optimal anisotropic meshes by predicting the desired element aspect ratio. In [Pfaff et al., 2020], the sizing field is predicted by a neural network for adaptive remeshing along with the system dynamics.

## 3   Method

### 3.1   Problem Statement

For each learning task, we consider a class of PDE problems $\mathbb{P}$ defined on the domain $\Omega$ in $D$ dimensions. Each specific sample $\mathcal{P}_n \in \mathbb{P}$ $(n = 1, 2, 3, \dots)$ is determined by PDE related information, such as source terms, boundary conditions, PDE parameters, solution fields, etc. We use $\mathcal{X}_n$ to parameterize such information, which will be fed into the neural network as raw features, so that the trained model can generalize to a class of PDEs. The sample-specific information $\mathcal{X}_n = \{\boldsymbol{p}_n, \mathcal{M}_n\}$ can be further categorized into global parameters $\boldsymbol{p}_n$ and position-dependent parameters $\mathcal{M}_n$. Depending on the type of the PDE problem and the terms we expect the trained model to be able to generalize to, some examples of $\boldsymbol{p}_n$ are wave number of the Helmholtz equation and viscosity coefficient of the Burgers' equation, and options of $\mathcal{M}_n$ are source terms, system states in the previous timestep for time-dependent equations, etc.

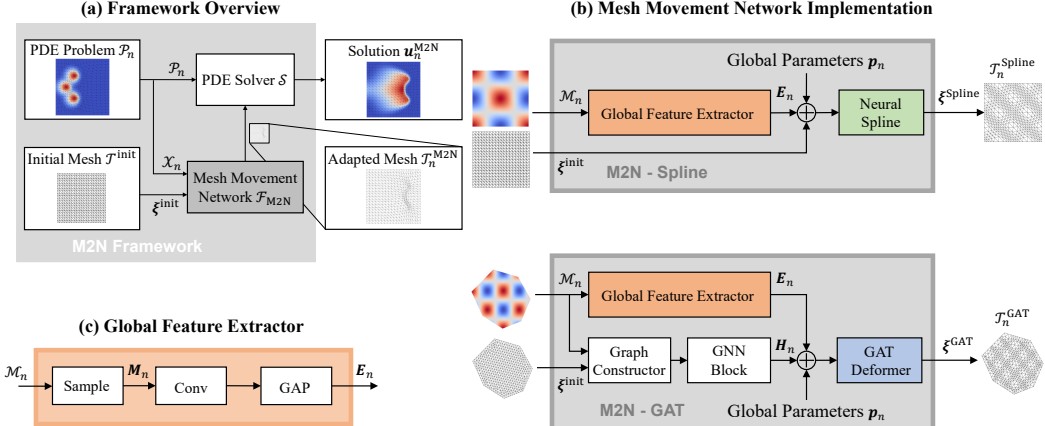

Figure 1: Proposed mesh movement network framework. (a) Given an initial mesh and an input state, the mesh deformer outputs an adapted mesh, which is then fed to the PDE solver. (b) The implementations of the Neural-Spline and the GAT based mesh movement networks. (c) The structure of the Global Feature Extractor.

Meshing is the procedure to spatially discretize a domain, which is necessary for most numerical methods to solve PDEs. Among these methods, the Finite Element Method (FEM) has been widely used in various engineering fields, which we will take as our solving method in the experiments. The initial mesh is generated with a traditional mesh generator. We denote the initial mesh by $\mathcal{T}^{\text{init}}$ and the mesh node positions in $\mathcal{T}^{\text{init}}$ by $\boldsymbol{\xi}^{\text{init}}$. Since the generation of the initial mesh is purely computational geometry based, the sample-specific information $\mathcal{X}_n$ cannot be utilized, hence $\mathcal{T}^{\text{init}}$ is shared by all samples $\mathcal{P}_n \in \mathbb{P}$. A standard PDE solver $\mathcal{S}$ will take as input the PDE problem $\mathcal{P}_n$ discretized on the mesh $\mathcal{T}^{\text{init}}$ and output the corresponding solution $\boldsymbol{u}_n^{\text{init}} = \mathcal{S}(\mathcal{P}_n(\mathcal{T}^{\text{init}}))$.

A high-quality adapted mesh can significantly improve the accuracy-efficiency trade-off of the PDE numerical solution. There are in general two types of mesh adaptation techniques. In this work, we consider mesh movement, or $r$-adaptation, which means we will not modify the number and topology of mesh nodes, but only relocate them. This characteristic provides the advantage that no remeshing needs to be performed, and there will be less repetitive computation for the PDE solver since the input matrices will maintain a constant size and sparsity structure [Budd et al., 2009]. A typical mesh movement method $\mathcal{F}$ will map the nodes $\boldsymbol{\xi}^{\text{init}}$ in the initial mesh $\mathcal{T}^{\text{init}}$ to $\boldsymbol{\xi}_n^{\text{adap}} = \mathcal{F}(\boldsymbol{\xi}^{\text{init}}, \mathcal{X}_n)$, so that the adapted mesh $\mathcal{T}_n^{\text{adap}}$ will be constructed with $\boldsymbol{\xi}_n^{\text{adap}}$ and the topology defined on the initial mesh. The PDE solution corresponding to the adapted mesh discretization $\boldsymbol{u}_n^{\text{adap}} = \mathcal{S}(\mathcal{P}_n(\mathcal{T}_n^{\text{adap}}))$ is expected to be much more accurate than $\boldsymbol{u}_n^{\text{init}}$, although the scale of the discretized problem that solver $\mathcal{S}$ receives is exactly the same. One of the most advanced mesh movement methods is the Monge-Ampère (MA) method. The mesh nodes optimized by the MA method $\boldsymbol{\xi}_n^{\text{MA}} = \mathcal{F}_{\text{MA}}(\boldsymbol{\xi}^{\text{init}}, \mathcal{X}_n)$ will be taken as the supervised signal to train the neural network model, while the computational efficiency and accuracy of the corresponding solution $\boldsymbol{u}_n^{\text{MA}} = \mathcal{S}(\mathcal{P}_n(\mathcal{T}_n^{\text{MA}}))$ will be compared against.

### 3.2 Framework Overview

The computational cost of sophisticated traditional mesh movement methods is often too expensive. In some cases, the cost of mesh adaptation is comparable to or even higher than that of solving the underlying PDE problem, which is generally unacceptable. Therefore, our goal is to model a Mesh Movement Network (M2N) based mapping $\mathcal{F}_{\text{M2N}}(\cdot|\theta)$, where $\theta$ represents the trainable parameters, such that the mesh adaptation process can be greatly accelerated. The proposed framework is demonstrated in Figure 1(a). We consider constructing a learning-based mesh movement method that relocates the nodes $\boldsymbol{\xi}^{\text{init}}$ in the initial mesh given the input state $\mathcal{X}_n$:

$$\boldsymbol{\xi}_n^{\text{M2N}} = \mathcal{F}_{\text{M2N}}(\boldsymbol{\xi}^{\text{init}}, \mathcal{X}_n|\theta), \tag{1}$$

from which the adapted mesh $\mathcal{T}_n^{\text{M2N}}$ can be reconstructed based on the topology of the initial mesh. From our empirical study, we take $\ell^1$ loss between the model output and the adapted mesh nodes

obtained with the Monge-Ampère method:

$$L(\theta) = \sum_{\mathcal{P}_n \in \mathbb{P}_{\text{train}}} \left\| \boldsymbol{\xi}_n^{\text{M2N}} - \boldsymbol{\xi}_n^{\text{MA}} \right\|_1. \tag{2}$$

Under the M2N framework, we implemented two network structures: a Neural-Spline based network and a GAT based network, which are denoted as M2N-Spline and M2N-GAT, respectively. Both models are designed to be capable of alleviating mesh tangling, keeping boundary consistency, and generalizing to mesh with different resolutions, which are key requirements for mesh movement. Because of their inherent characteristics, the M2N-Spline model behaves better when large global mesh deformation is required, while M2N-GAT is able to handle irregularly shaped domains and can better learn delicate local deformation. The detailed model structures will be introduced in the following sections, and more specifics of the model implementations can be found in Appendix A.

### 3.3 Neural-Spline based Network

As demonstrated in Figure 1(b), our Neural-Spline based network is mainly composed of two parts, where the input information $\mathcal{X}_n = \{\boldsymbol{p}_n, \mathcal{M}_n\}$ will be separately fed into the model. The global feature extractor GFE$(\cdot)$ extracts features from the position-dependent parameters $\mathcal{M}_n$ to obtain the mesh resolution invariant embedding $\boldsymbol{E}_n$. $\boldsymbol{E}_n$ together with the global physical parameters $\boldsymbol{p}_n$ will then be fed into the neural spline deformer to control the mesh node relocation, i.e.,

$$\mathcal{F}_{\text{M2N}}^{\text{Spline}}(\boldsymbol{\xi}^{\text{init}}, \mathcal{X}_n) = \text{Spline}(\boldsymbol{\xi}^{\text{init}}, \text{GFE}(\mathcal{M}_n) \oplus \boldsymbol{p}_n), \tag{3}$$

where operator $\oplus$ represents tensor concatenation.

**Global Feature Extractor** As shown in Figure 1(c), the global feature extractor has three modules:

$$\text{GFE}(\mathcal{M}_n) = \text{GAP}(\text{Conv}(\text{Sample}(\mathcal{M}_n))). \tag{4}$$

In detail, we uniformly sample the input state $\mathcal{M}_n$ in the domain $\Omega$. If the domain is with an irregular boundary, sampling is performed inside its minimum bounding box and the values of the sampling points outside the domain boundary are set to zero. The sampled states are assembled as a state tensor $\boldsymbol{M}_n$. To keep the extracted feature invariant to the magnitude, the state tensor $\boldsymbol{M}_n$ is normalized by its maximum absolute value. After normalization, $\boldsymbol{M}_n$ is sent into the convolutional layers for feature extraction, whose output is further fed into a Global Average Pooling (GAP) layer [Lin et al., 2013] to obtain a mesh resolution invariant global embedding $\boldsymbol{E}_n$. The embedding $\boldsymbol{E}_n$ will then be concatenated with global physical parameters $\boldsymbol{p}_n$ to obtain $\boldsymbol{I}_n$, which is fed into the deformer.

**Neural-Spline based Deformer** Normalizing flow models [Kobyzev et al., 2020] are proposed to learn invertible mappings. Neural spline [Durkan et al., 2019], as a specific type of normalizing flows, transforms the input with a differentiable monotone rational-quadratic spline function RQS$(\cdot|\boldsymbol{K})$, where $\boldsymbol{K}$ represents the learnable anchor points that determines the invertible mapping. In our model, the neural spline deformer Spline$(\boldsymbol{\xi}^{\text{init}}, \boldsymbol{I}_n)$ is a stack of neural spline layers RQS$_d(\boldsymbol{\xi}^{(d)}|\boldsymbol{K}_d(\boldsymbol{I}_n, \boldsymbol{\xi}^{(-d)}))$, each of which transforms one dimension of the input node coordinates $\boldsymbol{\xi}^{(d)}$, whose anchor points $\boldsymbol{K}_d$ are parameterized by the input features $\boldsymbol{I}_n$ and the other dimensions of node coordinates $\boldsymbol{\xi}^{(-d)}$.

Since the neural spline model is guaranteed to be an invertible mapping, mesh tangling can be alleviated. A specialty of the neural spline model is that, the end points of the spline function are fixed, therefore the intervals of the input and the output can be kept unchanged. In our case, this property is utilized to preserve the mesh with a hypercubic boundary (e.g. a rectangle in the 2-D case). Moreover, since the neural spline learns a continuous mapping, it can naturally generalize to different mesh resolution input.

### 3.4 GAT based Network

Although the Neural-Spline based network is well-designed for meshes of hypercubic domains, it is difficult to extend to domains with more general boundaries. Therefore, we also propose a Graph Neural Network (GNN) based model, which can naturally handle irregular domains. As shown in

Figure 1(b), the model consists of a two-branch feature extractor and a Graph Attention Network (GAT, [Veličković et al., 2017]) based mesh deformer GAT($\cdot$). In this subsection, we omit the sample index $n$ when there is no misunderstanding, hence: $\mathcal{F}_{\text{M2N}}^{\text{GAT}}(\boldsymbol{\xi}^{\text{init}}, \mathcal{X}) = \text{GAT}(\boldsymbol{\xi}^{\text{init}}, \text{LFE}(\mathcal{M}), \text{GFE}(\mathcal{M}) \oplus \boldsymbol{p})$.

**Feature Extractor** The feature extractor of the GAT-based network consists of two branches. One is the global feature extractor same as Eq. (4), the output of which will also be concatenated with global physical parameters $\boldsymbol{p}$ to obtain $\boldsymbol{I}$. The other is a GNN-based local feature extractor LFE($\cdot$). Our experiments empirically show that both branches are necessary for end-to-end performance.

For the local feature extractor, we need to first construct the input graph $G = (V, E)$ for each sample. The graph shares the same node number and graph topology with the initial mesh $\mathcal{T}^{\text{init}}$. For each graph node index $i \in \{1, \ldots, |V|\}$, the input feature $\boldsymbol{v}_i^{(0)}$ is the sampling of the input state $\mathcal{M}$ at $\boldsymbol{\xi}_i^{\text{init}}$, the position of node $i$ in the initial mesh $\mathcal{T}^{\text{init}}$. To preserve the mesh density information, the node distance $\|\boldsymbol{\xi}_i^{\text{init}} - \boldsymbol{\xi}_j^{\text{init}}\|$ are encoded into edge features $\boldsymbol{e}_{ij}^{(0)}$ as the relative edge features. The constructed graph will then be processed by the GNN block with a message passing mechanism to propagate the local physical information across the graph. The edge features are updated by a Multi-Layer Perceptron (MLP) $f_k(\cdot)$: $\boldsymbol{e}_{ij}^{(k)} \leftarrow f_k(\boldsymbol{e}_{ij}^{(k-1)}, \boldsymbol{v}_i^{(k-1)}, \boldsymbol{v}_j^{(k-1)})$, and the node features are updated by summing up the surrounding edge features $\boldsymbol{v}_i^{(k)} \leftarrow \sum_{j \in \mathcal{N}(i)} \boldsymbol{e}_{ij}^{(k)}$, where $k \in \{1, 2, \ldots K_{\text{GNN}}\}$ means the layer index, and $\mathcal{N}(i)$ refers to the neighbors of node $i$. At the output of the final layer, the features of each graph node are concatenated with the extracted global feature $\boldsymbol{I}$ to assemble $\boldsymbol{H}_i = \boldsymbol{v}_i^{(K_{\text{GNN}})} \oplus \boldsymbol{I}$. We use $\boldsymbol{H} = [\boldsymbol{H}_1, \ldots, \boldsymbol{H}_{|V|}]$ to represent the extracted features of the entire graph.

**GAT-based Deformer** As shown in Figure 2, the deformer consists of $K_{\text{GAT}}$ GAT blocks. The $k$-th block takes two inputs, mesh node positions $\boldsymbol{\xi}^{(k-1)}$ and extracted features $\boldsymbol{H}^{(k-1)}$, where $\boldsymbol{H}^{(k-1)} = \hat{\boldsymbol{H}}^{(k-1)} \oplus \boldsymbol{\xi}^{(k-1)}$. For the first block, $\hat{\boldsymbol{H}}^{(0)} = \boldsymbol{H}$ and $\boldsymbol{\xi}^{(0)} = \boldsymbol{\xi}^{\text{init}}$. The GAT block performs the following transforms:

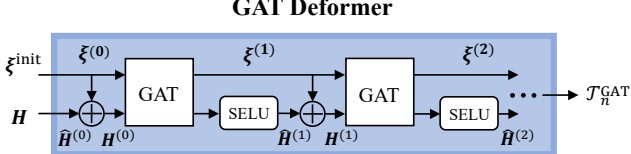

Figure 2: The implementation of the GAT Deformer.

$$\hat{\boldsymbol{H}}_i^{(k)} = \text{SELU}\left(\sum_{j \in \mathcal{N}_i} \alpha_{ij}^{(k)} \mathbf{W}^{(k)} \boldsymbol{H}_j^{(k-1)}\right), \qquad \boldsymbol{\xi}_i^{(k)} = \sum_{j \in \mathcal{N}_i} \alpha_{ij}^{(k)} \boldsymbol{\xi}_j^{(k-1)}, \tag{5}$$

where $\alpha_{ij}^{(k)}$ is the attention score indicating the importance of node $j$ to node $i$ in the $k$-th block, acquired by the method introduced in [Veličković et al., 2017], and $\mathbf{W}^{(k)}$ is a learnable weight matrix. To keep the mesh shape consistent before and after deformation, the nodes originally on the boundary are restricted to move along the boundary of the domain $\partial\Omega$. The adapted mesh $\mathcal{T}^{\text{GAT}}$ will be constructed from the output mesh node positions of the last block $\boldsymbol{\xi}^{(K_{\text{GAT}})}$. Because of the attention mechanism, during the mesh vertex relocation process, the movement of each mesh node is confined inside the convex hull composed of its 1-ring neighbors, hence effectively alleviating mesh tangling.

## 4 Experiment

We evaluate our proposed models against two learning-based baseline models and the traditional Monge-Ampère (MA) method, using error reduction ratio compared to the numerical solution on the initial mesh, mesh adaptation time consumption, and element inversion ratio, to validate their performance, generalization capability, and robustness. The experiments are conducted on the stationary and linear Poisson's equation in both a square domain and an irregular heptagonal domain, and the time-dependent non-linear Burgers' equation, where the supervised optimized meshes are generated with the traditional MA method. Each experiment is run three times with different random seeds to ensure the reliability of the model performances and provide mean and standard deviation of the results, which are summarized in the tables. More details of the experimental setup, dataset generation, model training, and experimental results can be found in the Appendix.

Table 1: Performance summary of the Poisson's equation problem on the square domain.

| Method | Error Reduction (%) | Time (ms) | Element Inversion (%) |
|---|---|---|---|
| MA (traditional) | 23.11 | 5220.99 | 0.00 |
| M2N-Spline | **20.82 ± 0.35** | 5.55 ± 0.01 | **0.00** |
| MLP-Deform-Clip | 16.74 ± 0.90 | **3.02 ± 0.03** | 1.60 |
| M2N-GAT | **20.38 ± 0.51** | 9.09 ± 0.02 | **0.00** |
| GAT-Deform-Clip | 19.95 ± 0.38 | 10.41 ± 0.05 | 3.11 |

As mentioned in Section 1, there is no similar previous work to compare results against. Comparisons with learning-based $h$-adaptation [Pfaff et al., 2020, Zhang et al., 2020, Fidkowski and Chen, 2021, Huang et al., 2021] will not be fair, because they belong to a different type of mesh adaptation methods, and the downstream PDE solver will receive discretized problems with different topology and scale. Therefore, we compare our proposed models with two baselines that can be interpreted as ablated versions of our proposed models. The model MLP-Deform-Clip replaces the neural-spline block with MLP. The model GAT-Deform-Clip replaces the GAT-deformer with the ordinary GAT block. Instead of learning the positions of mesh nodes, we set the learning target as the mesh nodes displacement for the baseline models, because it gives better performance according to experimental results. In order to enforce that baseline models can also preserve boundary consistency, the nodes moved out of the boundary will be pulled back into the domain, and the displacement component perpendicular to the boundary is clipped for the nodes which are supposed to stay on the boundary, as shown in Figure 3. However, there is no straightforward way to alleviate mesh tangling.

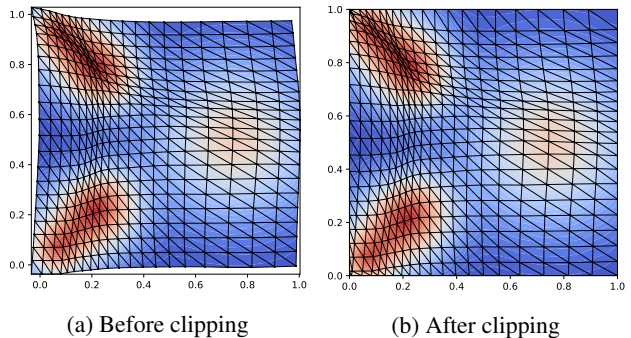

(a) Before clipping      (b) After clipping

Figure 3: Illustration of the mesh movement results of the baseline models before and after clipping.

## 4.1 Poisson's Equation

Poisson's equation is a second-order, linear, stationary PDE, which is widely used in electrostatics and thermodynamics, amongst other fields. We consider solving a class of 2-D Poisson's equations with different Dirichlet boundary conditions and mesh resolutions:

$$-\nabla \cdot \nabla u(x,y) = f(x,y), \quad (x,y) \in \Omega,$$
$$u(x,y) = u_0(x,y), \quad (x,y) \in \partial\Omega. \tag{6}$$

For both the square and heptagonal domain experiments, we generate analytical $u$ samples from a mixed Gaussian distribution, which are fed into Poisson's equation to obtain the corresponding source terms $f$ and boundary conditions $u_0$ as the problem samples, whereas the $u$ functions serve as the ground truth.

**Square Domain** In this experiment, we train the models on cases with mesh resolution of $15 \times 15$ and $20 \times 20$, each with 275 samples. To evaluate the models and how well they generalize to different mesh resolutions, we test on cases with mesh resolution from $12 \times 12$ to $23 \times 23$, each with 125 samples. Moreover, we deliberately set the optimal mesh movement to be drastic, in order to test how well different models can handle mesh tangling.

The quantitative results are summarized in Table 1. The proposed models, M2N-Spline and M2N-GAT, can achieve similar error reduction compared to the traditional MA method, while the mesh generation speed is two to three orders of magnitude faster. In addition, although the proposed models perform only marginally better than the two baseline models in error reduction and time, they are proven very effective to keep the mesh untangled. In comparison, the baseline models suffer from mesh tangling.

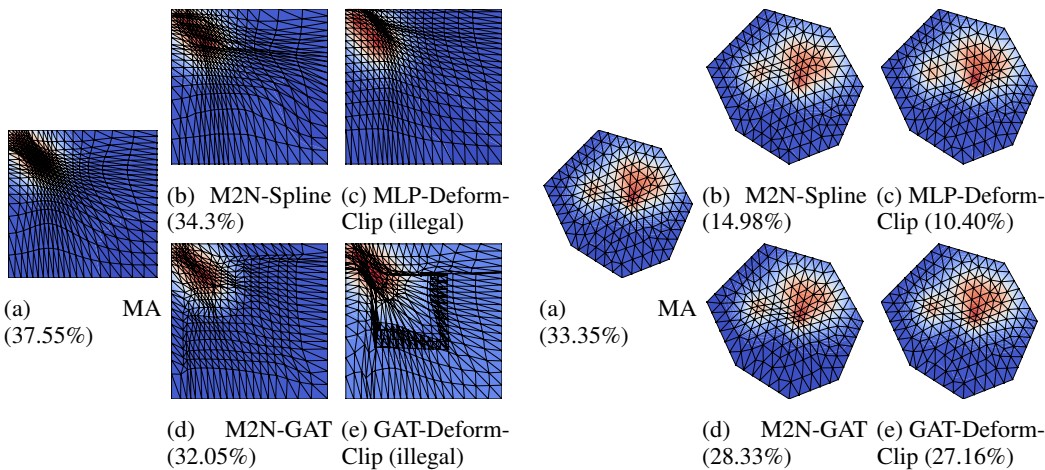

Figure 4: Comparison of mesh movement for an example problem of the Poisson's equation on the square domain. The percentage values in the parentheses are the error reduction ratios.

Figure 5: Comparison of mesh movement for an example problem of the Poisson's equation on the heptagonal domain. The percentage values in the parentheses are the error reduction ratios.

Table 2: Performance summary of the Poisson's equation problem on the heptagonal domain.

| Method | Error Reduction (%) | Time (ms) | Element Inversion (%) |
|---|---|---|---|
| MA (traditional) | 26.57 | 7329.78 | 0.00 |
| M2N-Spline | $16.15 \pm 0.40$ | $5.97 \pm 0.05$ | 0.28 |
| MLP-Deform-Clip | $16.49 \pm 0.46$ | $\mathbf{2.60 \pm 0.02}$ | 0.56 |
| M2N-GAT | $\mathbf{22.39 \pm 0.27}$ | $9.41 \pm 0.03$ | $\mathbf{0.00}$ |
| GAT-Deform-Clip | $\mathbf{22.40 \pm 0.47}$ | $10.81 \pm 0.07$ | 0.68 |

An example is given in Figure 4, where the mesh density in the upper left corner needs to be very high (shown in Figure 4(a)). It is demonstrated that M2N-Spline is more flexible for the cases where the overall mesh deformation is required to be large. On the other hand, for M2N-GAT, because of the constrained movement in each layer to alleviate mesh tangling and the finite layer numbers, it does not learn as well in such scenarios.

**Irregular Heptagonal Domain**  To evaluate the performance of different models for more general domain shapes, we conduct an experiment using Poisson's equation in an irregular heptagonal domain. The models are trained at mesh densities of 13, 16, 19, and 22, each with 320 samples, and tested on mesh densities from 12 to 23, each with 80 samples, to evaluate the performance and generalization capability of the models. The initial mesh $\mathcal{T}^{\text{init}}$ is generated with the Delaunay triangulation method provided by Gmsh [Geuzaine and Remacle, 2009].

The results are summarized in Table 2. It can be seen that all deep learning models can perform mesh movement around two to three orders of magnitude faster than the traditional MA method. Two GNN-based models perform better than the other two models, with the error reduction ratio comparable to the MA method. This is because the GNN-based models can naturally embed the information of the entire irregular domain into the network, and there are extra local feature extractors in the model. On the contrary, both the M2N-Spline and the MLP-Deform-Clip models are point-to-point mappings with only the global feature extractor, hence lacking such capabilities. The results of all methods for an example are shown in Figure 5, from which we can see that the GNN-based models can better capture the delicate local structure where mesh resolution needs to be increased. Although mesh tangling still will not occur for M2N-GAT, it happens for M2N-Spline, because for the M2N-Spline model, its guarantee only works for hypercubic boundaries (rectangle in the 2-D case).

Table 3: Performance summary of the Burgers' equation problem.

| Method | Error Reduction (%) | Time (ms) | Element Inversion (%) |
|---|---|---|---|
| MA (traditional) | 60.24 | 81590.64 | 0.00 |
| M2N-Spline | 48.92 ± 1.33 | 5.54 ± 0.02 | **0.00** |
| MLP-Deform-Clip | 43.53 ± 1.91 | **2.92 ± 0.01** | 0.00 |
| M2N-GAT | **57.75 ± 0.68** | 8.93 ± 0.01 | 0.00 |
| GAT-Deform-Clip | 51.69 ± 4.01 | 10.41 ± 0.01 | 0.53 |

## 4.2 Burgers' Equation

The viscous Burgers' equation is a non-linear, time-dependent PDE describing advection and diffusion processes in fluids. In this experiment, we consider a class of 2-D Burgers' equations with different previous states, viscosity coefficients, and mesh resolutions:

$$\frac{\partial u}{\partial t} + (u \cdot \nabla)u - \nu \nabla^2 u = 0, \quad (x, y) \in \Omega,$$
$$(n \cdot \nabla)u = 0, \quad (x, y) \in \partial\Omega, \tag{7}$$

where constant scalar $\nu > 0$ represents the viscosity coefficient and $u$ is the velocity vector field obeying this PDE. We define the problem on the unit square domain $\Omega = [0, 1]^2$.

In this experiment, we still train the models on cases with mesh resolution of $15 \times 15$ and $20 \times 20$, each with 9 trajectories and 60 timesteps per trajectory, and with different viscosity coefficients. Since a unit square domain is considered in this experiment, the initial uniform structured mesh $\mathcal{T}^{\text{init}}$ can be easily obtained by interpolation. To evaluate the models and how well they can generalize to different mesh resolutions, we test on cases with mesh resolution from $11 \times 11$ to $24 \times 24$, each with 8 trajectories and 60 timesteps per trajectory.

The results are summarized in Table 3. It can be found that all learning-based methods are three to four orders of magnitude faster than the traditional MA method. The acceleration is about one order of magnitude larger than Poisson's equation experiments, because the MA method runs even slower for a nonlinear PDE. Mesh tangling never occurs for M2N-Spline and M2N-GAT in this experiment. The GNN-based models perform better than the other two models, because they contain extra local feature extractors, and local information can be propagated better through edges, which is important for the Burgers' equation problem. Some generated mesh examples at different timesteps of four

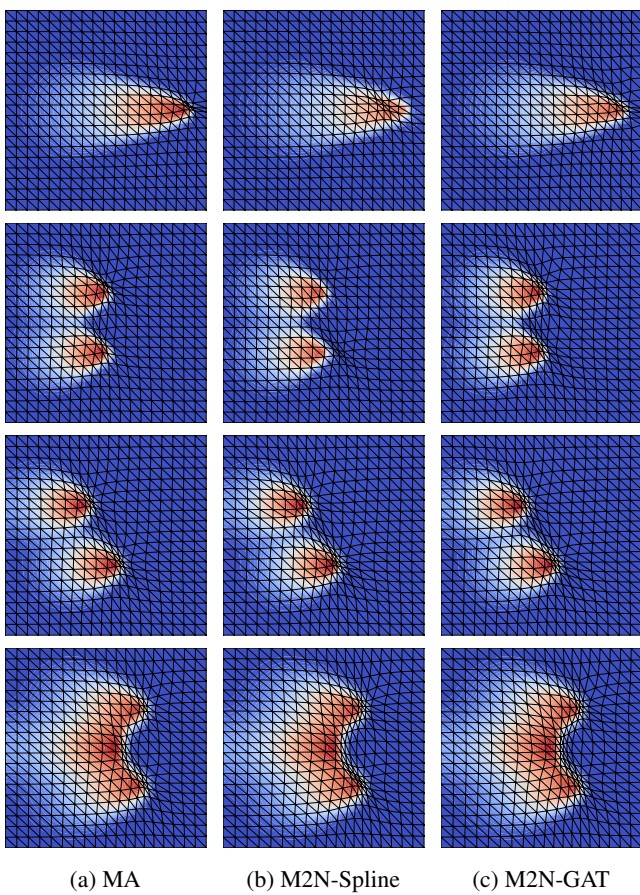

(a) MA      (b) M2N-Spline      (c) M2N-GAT

Figure 6: Comparison of mesh movement for the Burgers' equation problem. In each row is a different sample.

different trajectories are shown in Figure 6, where it can be seen that M2N-GAT is better at performing delicate local deformation compared to the M2N-Spline model.

## 5 Conclusion

In this paper, we have proposed the Mesh Movement Network (M2N), which to the best of our knowledge is the first learning-based end-to-end mesh movement method for PDE solvers. Traditional mesh movement methods can improve the accuracy of numerical PDE solutions without modifying the topology of the mesh, at the expense of solving an auxiliary PDE, which is often very computationally expensive and sometimes makes the approach infeasible. With the power of deep learning, M2N generates adapted meshes for different PDE problems of the same type, with the solution precision comparable to ground truth but at a much faster speed. To achieve this robustly, we have designed a Neural-Spline based and a GAT based mesh deformer, to guarantee the output adapted mesh retains boundary consistency, alleviates mesh tangling, and generalizes to different mesh densities. The results are validated on the static linear Poisson's equation with regular and irregular domains, and the time-dependent nonlinear Burgers' equation.

On the other hand, there are still several limitations and future directions worth discussing. First of all, although the proposed Neural-Spline based and GAT based models can effectively alleviate the mesh tangling issue, they cannot theoretically guarantee that. Therefore, analyses of mesh tangling avoidance and error reduction improvement can be conducted. Secondly, in this paper, we used the Monge-Ampère method to generate the supervised optimized meshes, while there are other traditional $r$-adaptation methods that can also be considered and tested. Finally, the scale and complexity of the current experiments are still far from practical applications. Experiments with more complicated boundary shapes, in larger scales, in higher dimensions, and for more diverse PDE types, can be conducted to better validate the proposed methods.

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
