# Appendix of
# M2N: Mesh Movement Networks for PDE Solvers

## A    Additional Model Details.

**Global Feature Extractor**    As mentioned in the paper, the global feature extractor GFE$(\cdot)$ is composed of 3 modules: GFE$(\mathcal{M}_n) = $ GAP(Conv(Sample$(\mathcal{M}_n)$)). The sampling module Sample$(\cdot)$ is implemented by the built-in interpolation interface in Firedrake[Rathgeber et al., 2016], with sampling density 32x32. The convolutional block contains 4 convolutional layers. The SELU [Klambauer et al., 2017] activation function is used to increase the representation capability of the model. The output tensor from the convolutional block is then fed into a Global Average Pooling (GAP) layer to get a mesh resolution invariant global feature embedding $\boldsymbol{E}_n$.

**Neural-Spline based Deformer**    The Neural-Spline based Deformer is implemented based on the open source code [Tony Duan, 2020]. In neural spline, the invertible mapping is determined by a differentiable monotone rational-quadratic spline function RQS$(\cdot|\boldsymbol{K})$, where $\boldsymbol{K}$ represents the learnable anchor points. As described in the paper, the neural spline deformer Spline$(\boldsymbol{\xi}^{\text{init}}, \boldsymbol{I}_n)$ is a stack of neural spline layers RQS$_d(\boldsymbol{\xi}^{(d)}|\boldsymbol{K}_d(\boldsymbol{I}_n, \boldsymbol{\xi}^{(-d)}))$. The anchor points $\boldsymbol{K}_d$ are parameterized by the input features $\boldsymbol{I}_n$ and the other dimensions of node coordinates $\boldsymbol{\xi}^{(-d)}$, which are represented by a SELU-activated six-layer MLP with output size of $(3a-1)$. Here $a$ is a hyper-parameter and is set to be 20 for all the experiments. The output vector $\boldsymbol{\theta}$ of the MLP can be partitioned as $\boldsymbol{\theta} = \left[\boldsymbol{\theta}^w, \boldsymbol{\theta}^h, \boldsymbol{\theta}^d\right]$, where $\boldsymbol{\theta}^w$ and $\boldsymbol{\theta}^h$ are with length $a$, and $\boldsymbol{\theta}^d$ is with length $(a-1)$. Vectors $\boldsymbol{\theta}^w$ and $\boldsymbol{\theta}^h$ are firstly normalized by a softmax layer and then used to decide the positions of $(a+1)$ anchor points. The vector $\boldsymbol{\theta}^d$ is passed through a softplus layer (to keep derivative always positive) and directly used as the derivatives of the inner $(a-1)$ anchor points. For the two end points, their derivatives are set to be constant. In one neural spline block, each dimension of input points takes turns to be transformed. In all experiments, only one neural-spline block is used.

**Feature Extractor of GAT-based network**    In the GAT-based network, an additional GNN-based local feature extractor LFE$(\cdot)$ is introduced to help capture local information, which is composed of one GNN-block, i.e., $K_{GNN} = 1$. For LFE$(\cdot)$, the input graph $G = (V, E)$ is constructed according to the process illustrated in Figure 8. The edge feature encoder $f_k(\cdot)$ of the GNN-block is implemented with a SELU-activated three-layer MLP with hidden layer size of 64 and output size of 16.

**GAT-based Deformer**    In all experiments, the GAT-based Deformer consists of 5 GAT blocks, i.e., $K_{GAT} = 5$. The layer sizes of each GAT block are 256, 256, 512, 256 and 20 respectively. For the $k$-th GAT block, it transforms the upstream mesh vertex positions $\boldsymbol{\xi}^{(k-1)} = [\boldsymbol{\xi}_1^{(k-1)}, ..., \boldsymbol{\xi}_{|V|}^{(k-1)}]$ to $\boldsymbol{\xi}^{(k)} = [\boldsymbol{\xi}_1^{(k)}, ..., \boldsymbol{\xi}_{|V|}^{(k)}]$, where $\boldsymbol{\xi}_i^{(k)} = \sum_{j \in \mathcal{N}_i} \alpha_{ij}^{(k)} \boldsymbol{\xi}_j^{(k-1)}$. The updating process is illustrated in Figure 7. In such a way, the new position of each mesh node is confined inside the convex hull composed of its 1-ring neighbors with previous positions, which can greatly alleviate mesh tangling. To keep the mesh nodes of irregular meshes always on the boundary, graph cutting is needed for the initial constructed graph. In Figure 8b, the initial graph is constructed by simply converting the initial mesh edges into bidirectional graph edges. However, in the processed graph in Figure 8c, graph edges

from interior nodes to boundary nodes (except the corner nodes), i.e., the pink nodes in Figure 8c, are removed from the initial graph. In this manner, the mesh nodes on the boundary will only move along the boundary without leaving it since their neighbor nodes connected by the incoming edges are all also on the boundaries. As for the corner mesh nodes, they are always fixed. Therefore, the interior mesh nodes will never move outside the boundary because each mesh node moves inside the convex hull composed of 1-ring neighbors of itself. Finally, boundary consistency is also achieved based on the above mechanism.

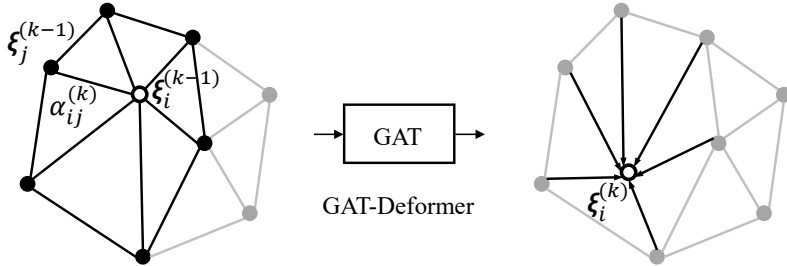

Figure 7: Illustration of the GAT based mesh deformer. The new position of each mesh node is confined inside the convex hull composed of its 1-ring neighbors with previous positions.

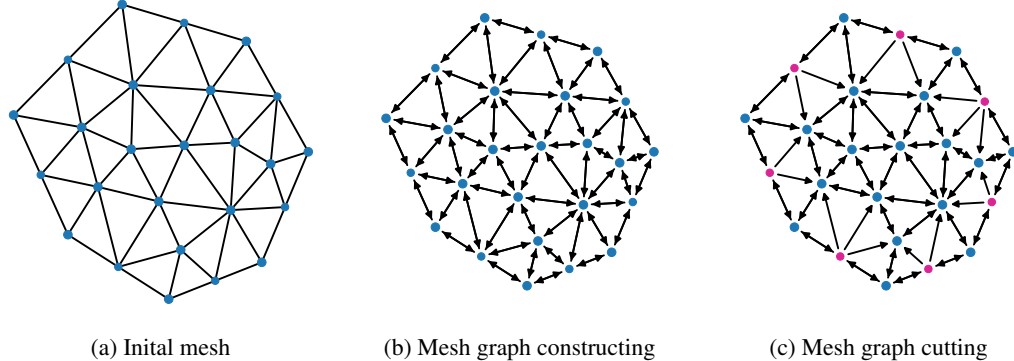

(a) Inital mesh      (b) Mesh graph constructing      (c) Mesh graph cutting

Figure 8: Illustration of graph construction process. (a) The initial mesh, where blue nodes are mesh nodes and black edges are mesh edges. (b) The initial mesh is converted to a graph, where blue nodes are graph nodes and black edges are bidirectional graph edges. (c) The pink nodes are boundary nodes but not at corners. The edges starting from interior blue graph nodes to pink graph nodes are removed from the graph.

**Baselines** To ensure the fairness of the comparison between our proposed two models and their corresponding baselines, the scale of the networks is similar except for the last layer. In the two baseline models MLP-Deform-Clip and GAT-Deform-Clip, the sizes of the last layer are set as 2 to predict the displacement of each mesh node. It should be noted that the predicted displacement of corner mesh nodes are always set as 0. Furthermore, to preserve consistency of the domain, the displacement of boundary nodes are projected onto the domain boundary. This means that such nodes can only move tangentially to the boundaries. In some cases, because there is no constraint on the predicted displacements output by the naive MLP and GAT, some interior mesh nodes can be located outside of the boundaries after adding the predicted displacements. Under such circumstances, these outside mesh nodes are directly projected onto the nearest boundaries. However, this will cause mesh tangling.

## B    Experimental Setup

**Software.** For all experiments, we use Firedrake [Rathgeber et al., 2016], a Python library for automating the numerical solution of PDEs. We implement our models with Py-

Torch [Paszke et al., 2019], and use PyTorch Geometric (PyG) [Fey and Lenssen, 2019] for graph neural networks. The code for generating the target meshes are based on the repository by Wallwork [2022]. A MindSpore version of the code will be available at https://gitee.com/mindspore/models/tree/master/research/hpc/m2n.

**Monitor Function.** In all experiments, we use the Monge-Ampère (MA) method to generate the target deformed mesh. For the MA method, the mesh is equidistributed with respect to a user-defined monitor function. The monitor function provides a concept of 'mesh density' and controls the sense in which errors are equidistributed by the mesh adaptation algorithm. Therefore, the choice of monitor function greatly impacts the geometry of the deformed mesh and should be chosen with care. In this paper, we define the monitor function as

$$m = 1 + \alpha \frac{(u - u_{exact})^2}{\max_{x,y}(u - u_{exact})^2} + \beta \frac{\|H(u)\|_F}{\max_{x,y}\|H(u)\|_F}, \tag{1}$$

where $u$ is the numerical solution on the current mesh, $u_{exact}$ is the numerical solution on a uniformly refined mesh or analytical solution, $\|H(u)\|_F$ represents the Frobenius norm of the Hessian matrix of the solution $u$, and $\alpha$ and $\beta$ are two non-negative coefficients, which weight the importance of the two terms.

**Evaluation Methods.** We evaluate the performance of different models from three aspects: error reduction ratio, mesh generation speed and tangling avoidance.

The error reduction ratio is calculated by the formula $(e_{\text{initial}} - e_{\text{adapted}})/e_{\text{initial}}$, where $e_{\text{initial}}$ and $e_{\text{adapted}}$ are the $\ell^2$ norms of the discretization errors of the PDE solutions on the initial mesh and the deformed mesh compared with the ground truth respectively. The ground truth of the PDE solution is provided either with the analytical solutions or the numerical solutions on meshes with very high resolution. In the experiments of Poisson's equation problems, the ground truth is provided by analytical solutions, while in the experiment of Burgers' equation problems, the ground truth is provided by the numerical solution on a fine mesh with the resolution of $100 \times 100$.

We assess whether a particular model leads to mesh tangling by analyzing the Jacobian $J_K$ of the affine transformation between the reference element $\widehat{K}$ and a particular element $K$ of the adapted mesh:

$$F : \widehat{K} \to K, \qquad J_K = \nabla F. \tag{2}$$

Initially the sign of $\det(J_K)$ will be either positive or negative, depending on the mesh ordering used. If this determinant changes sign after a mesh adaptation step then this means that the element has been inverted, i.e., the mesh has tangled.

**Training Details.** In the experiments, the models are trained with $\ell^1$ loss function, and an Adam optimizer [Kingma and Ba, 2014] with learning rate of $10^{-3}$ on two RTX-2080ti GPUs. For the experiment of Poisson's equation problems in a square domain, it takes around 3 hours to train each of the two GNN-based models, namely M2N-GAT and GAT-Deform-Clip, while it takes around 1.5 hours to train each of the other two models, namely M2N-Spline and MLP-Deform-Clip. For the experiments of Poisson's equation problems on a heptagonal domain and Burgers' equation problems, the training time for each model is about twice than that of Poisson's equation problems on a square domain. To ensure the reliability of the results, each training is run three times with different random seeds.

**Testing Details.** The Monge-Ampère method is run on an 8-core i7-7820X CPU. The implementation of the Monge-Ampère method naturally supports parallelization since Firedrake itself supports MPI (Message Passing Interface), which uses PETSc and libblas-3.7.1 as the underlying linear algebra library. However, we only use single processing across the experiments, because we find that for our experiments, multiprocessing actually does not provide a performance boost. For the other learning-based models, we run experiments on one RTX-2080ti GPU.

## C    Dataset Generation

Some specifics of the dataset generation are summarized in Table 4. In total, three datasets are generated, which are Poisson's Equation problem on a Square domain, Poisson's Equation problem on a Heptagonal domain, and Burgers' Equation problem on a Square domain.

### C.1    Poisson's Equation

For the two datasets of Poisson's equation problems, the first degree continuous Lagrange polynomial is used as the basis function for the Finite Element Method. We generate a manufactured analytical solution $u$ by sampling from a mixed Gaussian distribution, which can be fed into Poisson's equation to obtain the corresponding source terms $f$ and boundary conditions $u_0$. The analytical solution $u$ follows the form of $u = \sum_{i=1}^{n} \exp\left(-\frac{(x-a_i)^2+(y-b_i)^2}{w_i}\right)$. The target optimized meshes are generated using the Monge-Ampère method.

**Poisson's Equation on an square domain**    In the square domain, we sample $n \in \{1, ..., 6\}$, $w_i \in [0.01, 0.2]$. The center of each Gaussian function is sampled inside the square domain. To train the model, 275 samples are generated under the each mesh resolution of $15 \times 15$ and $20 \times 20$. To test the generalization ability of the model, 125 samples are further sampled on each mesh resolution ranging from 12 to 23. Some examples are shown in Figure 9.

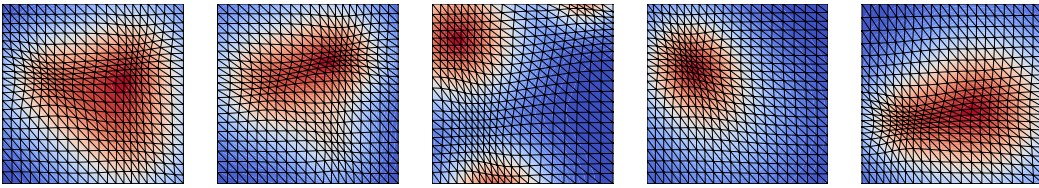

Figure 9: Some examples of Poisson's equation problem on the square domain.

**Poisson's Equation on an irregular heptagonal domain**    To generate an initial irregular heptagonal mesh, we use Gmsh [Geuzaine and Remacle, 2009], an open source finite element mesh generator. In the irregular heptagonal domain, we sample $n \in \{1, ..., 5\}$, $w_i \in [0.005, 0.01]$. The center of each Gaussian function is sampled inside the irregular domain. To construct the training set, 320 samples are generated under the each mesh density of 13, 16, 19, and 22. For the test set, 80 samples are generated for each mesh density from 12 to 23. We provide some examples shown in Figure 10.

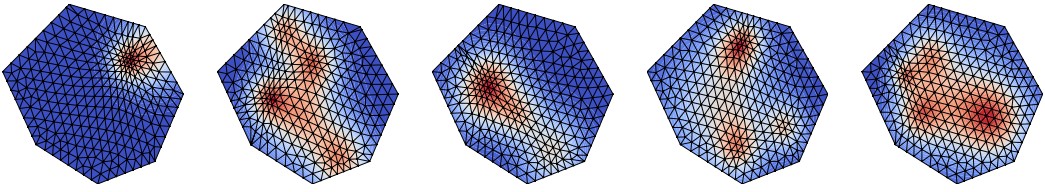

Figure 10: Some examples of Poisson's equation problem on the heptagonal domain.

### C.2    Burgers' Equation

In the Burgers' equation experiment, the second degree continuous Lagrange polynomial is chosen as the basis function. For simplicity and stability, a backward Euler scheme is used. The simulation time step is set as 1/30 second and each trajectory runs for 60 time steps. The initial condition is set as $\mathbf{u}_{\text{initial}} = [u_x, 0]$, where $u_x = \sum_{i=1}^{n} \exp\left(-\frac{(x-a_i)^2+(y-b_i)^2}{0.01}\right)$. We generate 9 trajectories for training on two mesh resolution of 15 and 20. These trajectories are generated by pairing three different initial conditions and three different viscosity coefficients $\{0.0001, 0.001, 0.002\}$. To evaluate the generalization ability of different models, 8 trajectories with unseen viscosity coefficients $\{0.005, 0.0015\}$ and initial conditions are generated for the mesh resolution from 11 to 24. The generation

time of Burgers' Equation dataset is much longer than the above two Poisson's equation datasets because of the nonlinearity of Burgers' equation and the time-consuming mesh-to-mesh projection step in Firedrake. Some examples of trajectories are provided in Figure 11.

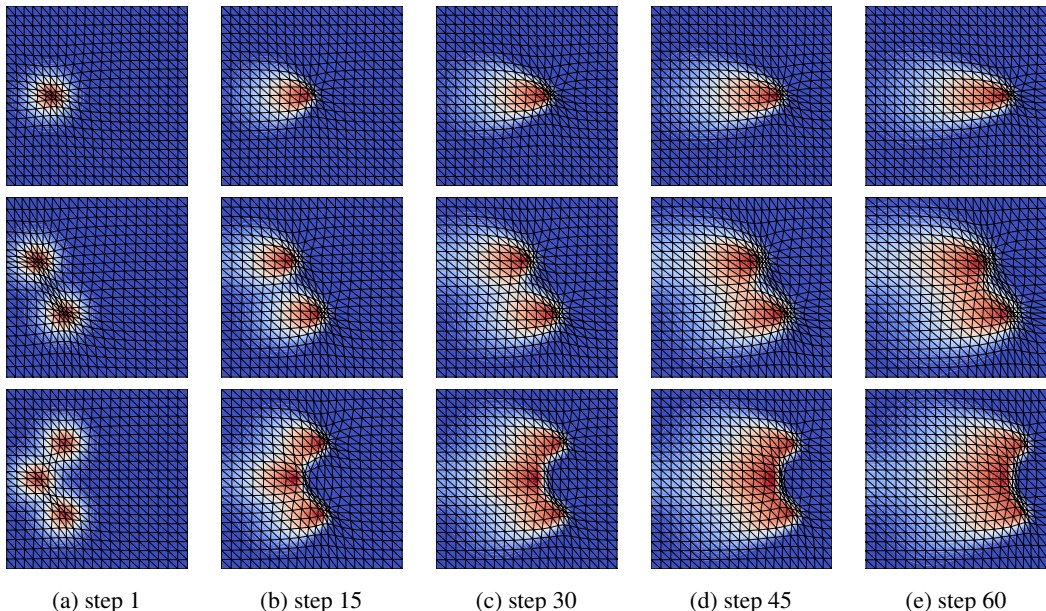

(a) step 1          (b) step 15          (c) step 30          (d) step 45          (e) step 60

Figure 11: Some examples of the trajectories of the Burgers' equation.

# D Extra Experiment Results

In Section 4 of the main paper, the error reduction ratio results summarized in Tables 1, 2 and 3 are averaged over the results under all mesh resolutions in the test set. Here, we provide the detailed error reduction ratio results under each mesh resolution for all the three experiments, namely Poisson's equation problem on the square domain, Poisson's equation problem on the heptagonal domain, and Burgers' equation problem, shown in Figure 12, 13 and 14, respectively.

By analyzing the results given in Figure 12, 13 and 14, several conclusions can be obtained.

First of all, the error reduction ratio gap between M2N-Spline and MLP-Deform-Clip is generally larger than that of M2N-GAT and GAT-Deform-Clip, which is because the spline module can simultaneously provide more representation power and robustness for mesh movement network compared with the vanilla MLP module, thus achieving higher error reduction ratio. However, we also need to point out that M2N-GAT and GAT-Deform-Clip have basically the same network size, where the only difference between them is the specially-designed attention-based anti-tangling mechanism for M2N-GAT. On the other hand, Spline-based and MLP-based deformers are of totally different network structures, hence cannot be totally fairly compared in scale.

Secondly, by comparing between Figure 13 and the other two figures, namely Figure 12 and 14, we can see that the error reduction ratio gap between four learning based methods and MA method in the experiment of Poisson's equation problem on the heptagonal domain is larger than that of other two experiments. This is because, compared with a square domain, the degree of mesh movement has a greater impact on the error reduction ratio in the case of a heptagonal domain. Therefore, the learning based methods need higher mesh movement accuracy, compared with the other two experiments, to achieve similar error reduction ratio with the MA method.

Thirdly, it can be seen from Figure 14 that the error reduction ratio of the MA method gradually arises with the increase of mesh resolution. This phenomenon is caused by the fact that the second degree continuous Lagrange polynomial is used as the basis function in the experiment of Burgers' equation problem. The utilization of higher order basis functions makes mesh adaptation more effective.

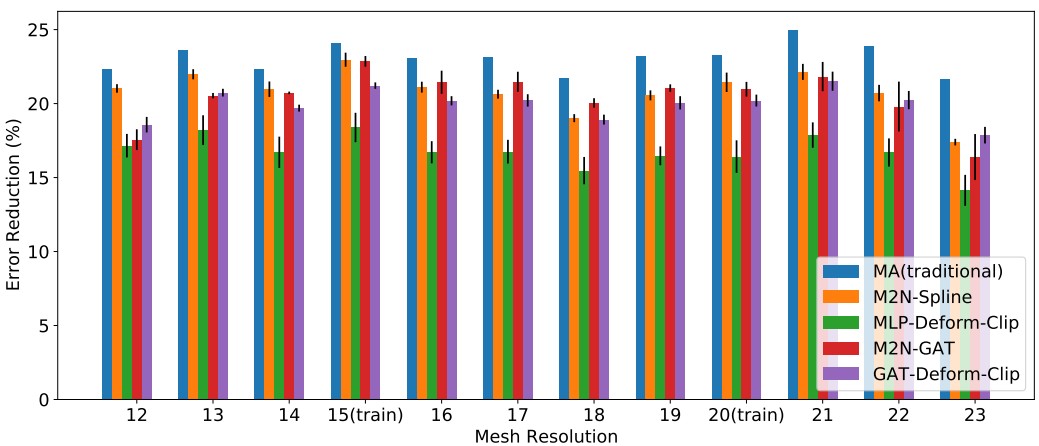

Figure 12: Error reduction ratio comparison under each mesh resolution of the Poisson's equation problem on the square domain.

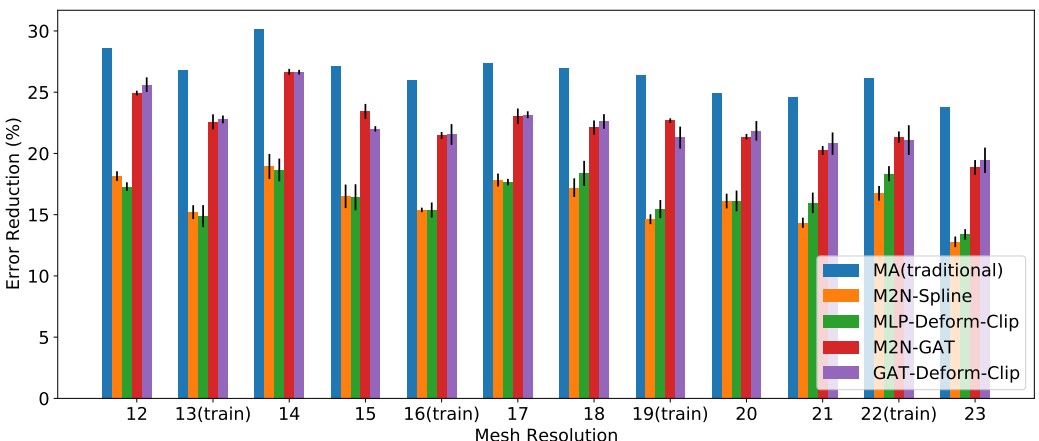

Figure 13: Error reduction ratio comparison under each mesh resolution of the Poisson's equation problem on the heptagonal domain.

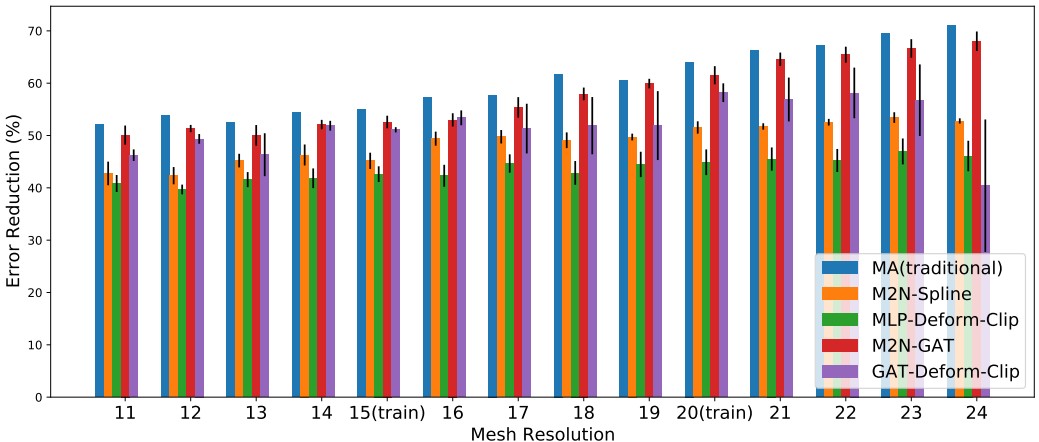

Figure 14: Error reduction ratio comparison under each mesh resolution of the Burgers' equation problem.

Table 4: Dataset description.

| Dataset | Domain | PDE Type | Construction Time (h) | Generalization Item | Resolution for Training | Resolution for Testing | Sample for Training per Mesh Resolution | Sample for Testing per Mesh Resolution | M.A. Coefficient |
|---|---|---|---|---|---|---|---|---|---|
| Poisson Square | Square | Linear Static | 0.5 | source term mesh density | 15, 20 | 12 - 23 | 275 | 125 | $\alpha = 0, \beta = 6$ |
| Poisson Heptagonal | Heptagonal | Linear Static | 1.0 | source term mesh density | 13, 16, 19, 22 | 12 - 23 | 320 | 80 | $\alpha = 6, \beta = 0$ |
| Burgers | Square | Nonlinear Time-Dependent | 9.0 | solution field physical parameter mesh density | 15, 20 | 11 - 24 | $9 \times 60$ | $8 \times 60$ | $\alpha = 0, \beta = 15$ |