# OpenReview forum: "M2N: Mesh Movement Networks for PDE Solvers"
_NeurIPS.cc/2022/Conference — NeurIPS 2022 Accept_

### Official Review · Reviewer_AXGX · 2022-07-11

**Rating:** 4
**Confidence:** 5
**Soundness:** 3 good
**Presentation:** 3 good
**Contribution:** 2 fair

**Summary:**

The authors have presented a novel approach for r-adaptivity based adaption for PDEs using two different network architectures. The Neural spline-based model serves well to avoid mesh tangling while handling meshes with large deformation, and the GAT-based model can generalize for domains with arbitrary shapes. The authors use the Monge-Ampere method to train using supervised learning, and demonstrate the effectiveness of their method on stationary and time-dependent, linear and non-linear equations, as well as regularly and irregularly shaped domains.

**Questions:**

(1) In irregular domains, the sampling is done by using a bounding box and values outside the actual domain are set to 0. We have found that such an approach is problematic if it used for solutions that have non-smooth (shocks) features. It would be good to note this limitation.
(2) Can the boundary nodes move with the neural spline model? Node movement along the boundary can be crucial for certain applications.
(3) Regarding the last two lines right before Section (4): Two adjacent nodes can move within the convex hull of their neighbors in such a way that the resulting mesh is tangled. Do you mean something else (e.g., half the distance from the closest neighbor)?

**Limitations:**

While this work is a step in the right direction, the proposed methods must really be demonstrated to work for large scale 2D/3D problems if they are to be claimed to be competitive against existing methods.

**Strengths And Weaknesses:**

Strength:
(1) The proposed framework is truly the first learning-based approach for r-adaptivity of meshes.
(2) The authors have considered two different approaches in Neural splines and GAT, and demonstrated their effectiveness along with their limitations.

Weaknesses:
The scope of the work seems somewhat limited in the sense of:
(1) the authors consider  only the Monge Ampere method for r-adaptivity. See "Simulation-driven optimization of high-order meshes in ALE hydrodynamics", "rp-adaptation for compressible flows", and "Defining a stretching and alignment aware quality measure for linear and curved 2D meshes", which have been used for r-adaptivity of meshes to PDEs.
(2) the problems considered are too simple; element count in each direction is ~20 and the simulations are 2D only.

---

> ### Author Response · Authors · 2022-08-02
> **Response to Reviewer AXGX**
>
> Thank you for your detailed and helpful comments and suggestions.
>
> - **Consider only the Monge-Ampère method:** thank you for the provided literature. We will definitely give them a detailed check and utilize them in our future work. As you mentioned in the review, the Monge-Ampère method is only used to generate data samples for supervised learning. Different r-adaptation methods don't have fundamental conflicts because they all pursue the error equidistribution principle. Therefore there should be no problem to replace the Monge-Ampère method with other advanced r-adaptation methods.
> - **Experiment scale:** we totally agree with you that the proposed methods need to work on large-scale problems to be really practical. On the other hand, we would also like to highlight that, this is to the best of our knowledge the first trial towards a learning-based end-to-end mesh movement method, and our experiments can serve as proof of concept to demonstrate its effectiveness.
> - **Sampling is done by using a bounding box in irregular domains:** thank you very much for sharing your valuable experience and the constructive suggestion. We also find the approach, though effective in our experiments, inelegant. That's why in the paper we list "evolution of the Neural-Spline based model to better handle irregular domains" as one of the future directions. One possible way is to perform the transfinite interpolation so that the neural spline model can still work in hypercubic domains but apply to irregular domain problems.
> - **Can the boundary nodes move with the neural spline model?** Yes, this is another advantage of the neural spline model, that it naturally supports the boundary nodes to move along the boundary without tangling. We would also like to point out that, our proposed GAT-based model also supports boundary nodes to move along the boundary, as explained in detail in Appendix A and Figure 2 in the Appendix.
> - **Last two lines right before Section 4:**
>     - Your understanding is accurate that introducing the GAT mechanism into the model can not theoretically avoid mesh tangling. That is why in the paper we only claim the proposed network structure can "alleviate mesh tangling", and we have listed "theoretical analysis on mesh tangling avoidance" as one of the future directions in the paper. On the other hand, we should note that, by keeping each vertex within the convex hull of its neighbors, mesh tangling can be greatly alleviated, which is empirically proven by the experiments.
>     - It is very observant of you to propose "half the distance from the closest neighbor". We have thought about it too, but sadly we found counter-examples that even in this case, mesh tangling can still happen. Since it will heavily reduce the flexibility of vertex movement, we finally decided to give up such mechanism.

---

### Official Review · Reviewer_h6GN · 2022-07-12

**Rating:** 7
**Confidence:** 3
**Soundness:** 3 good
**Presentation:** 3 good
**Contribution:** 3 good

**Summary:**

The paper presents two neural architectures for warping meshes to improve finite element modeling performance (reduce error). The models are trained in a supervised manner to match the maps produced by a Monge-Ampere (MA) mesh movement method. Error reduction is comparable to the classical MA method, but with a speed improvement of several orders of magnitude.

**Questions:**

The examples shown in the paper are on relatively coarse, high-quality meshes of domains with convex boundaries. How does the method perform on larger meshes, or when the initial mesh is less regular? Can it be applied to other types of PDE?

**Limitations:**

1. The models are only tested on two PDEs on simple domains. It would be good to see at least a few more evaluations, especially on larger/finer-grained meshes.
2. Even though comparison to h-adaptation methods may not be useful to evaluate the performance of this method, it might still be useful for practitioners to choose which type of adaptation to use.

**Strengths And Weaknesses:**

The overall idea of using a normalizing flow to move a mesh around seems sound, and allows for a nice tradeoff between the speed advantages of neural methods and the reliability of traditional FEM. The neural spline model can only natively preserve boundaries that are aligned to the axes. Also, while the neural spline model may produce a diffeomorphism, this does not guarantee avoiding mesh tangling because the discrete mesh might not be able to represent a sufficiently complicated smooth diffeomorphism. For the graph attention model, the use of neighborhood attention to keep vertices within the convex hulls of their neighbors is a nice observation, and could potentially guarantee injectivity under suitable conditions (as with Tutte embedding).

---

> ### Author Response · Authors · 2022-08-02
> **Response to Reviewer h6GN**
>
> Thank you for your detailed and helpful comments and suggestions.
>
> - **Limits of the neural-spline based model:**
>     - In our current version, as you mentioned, the neural spline model can only natively preserve boundaries that are aligned to the axes. But we would also like to point out that, the method holds the potential to generalize to irregular domains, for example by performing the transfinite interpolation.
>     - It is very observant of you to realize that a diffeomorphism does not necessarily guarantee to avoid mesh tangling in the discrete space. We have realized the same thing and also constructed a counter-example. That is why in the paper we only claim the proposed network structure can "alleviate mesh tangling". On the other hand, we should also realize that, for most practical cases, mesh tangling will be avoided by the current design, and the experiments empirically prove that. We are also pursuing to provide sufficient (and necessary) conditions, and we have listed "theoretical analysis on mesh tangling avoidance" as one of the future directions in the paper.
> - **Experiment scale:** we totally agree with you that the proposed methods need to work on large-scale problems to be really practical. On the other hand, we would also like to highlight that, this is to the best of our knowledge the first trial towards a learning-based end-to-end mesh movement method, and our experiments can serve as proof of concept to demonstrate its effectiveness.
> - **Applicability to other types of PDE:** in this paper, the two types of PDE we considered are quite different, i.e., the stationary and linear Poisson’s equation, and the time-dependent non-linear Burgers’ equation. Moreover, we train the network with supervised learning, which means the neural network is facing tasks with similar difficulties, regardless of the underlying PDE. Therefore we think it is convincing that the proposed method can be applied to other types of PDE.
> - **h-adaptation:** thank you for the suggestion. This is actually a very interesting topic which we also have repeatedly thought over. Here is something we would like to share and discuss with you.
>     - First of all, we agree that for end-users, they should hold the freedom to choose whichever kind of mesh adaptation, including h-adaptation, r-adaptation, or even mixed rh-adaptation, except for certain scenarios where the mesh topology has to be kept unchanged.
>     - As cited in the paper, there have been several pioneering works on learning-based h-adaptation. They are all not end-to-end, by which we mean the neural network only generates certain local metrics, which will then be fed into a traditional mesher/remesher to finally obtain the adapted mesh. We believe it is because, modifying the topology of a mesh, as a fundamentally discrete operation, is not something deep learning is good at.

---

### Official Review · Reviewer_2ifw · 2022-07-12

**Rating:** 4
**Confidence:** 4
**Soundness:** 2 fair
**Presentation:** 3 good
**Contribution:** 2 fair

**Summary:**

The paper proposes learning-based mesh movement methods for improving the accuracy of PDE solutions. The key idea is to train neural networks to approximate the solutions generated using the traditional Monge-Ampère method. The models are evaluated using two types of PDE problems on different domains, where the proposed model achieved performance close to the traditional method while being much faster.

The main contribution of the paper is the proposed neural models that can quickly produce a mesh movement result close to the traditional Monge-Ampère method.

**Questions:**

1) What type of CPU were the experiments run on? Are the implementation of the MA method single-threaded or paralellized?

2) What types of meshes are good for PDE solving? Why is not possible to encode these quality meshes into the loss function?

3) How do the models perform on large meshes and non-planar domains?

**Limitations:**

The discussion is sufficient.

**Strengths And Weaknesses:**

Strength:
- The proposed models can achieve performance close to the traditional Monge-Ampère method while being much faster.
- The models are evaluated on two types of PDE problems to show their performance.

Weakness:
- According to the literature cited in the paper, the Monge-Ampère method has been applied to non-planar domains such as spheres. But the evaluation in this paper is only performed on planar domains. It is unclear how well the method performs on domains with non-zero Gaussian curvature.
- Whilst the speed-up reported in the paper is impressive, they can also be misleading. The traditional MA method is run on the CPU, whilst the proposed models are run on the GPU. In the paper there is no information on the CPU model. Moreover, it is unclear what types of underlying linear algebra library the Firedrake code uses (NumPy can use different underlying BLAS libraries, which may have very different performance). So it is difficult to assess the actual speedup.
- The proposed models are formulated purely based on closeness to the Monge-Ampère solution. However, Monge-Ampère is not the ultimate goal of the mesh movement problem, but rather just one particular way to generate high-quality meshes. The paper does not provide much information on what types of meshes are good for PDE solving, which makes it difficult to judge the quality of the result beyond its similarity to the Monge-Ampère solution. This somewhat limits the novelty of the proposed approaches.
- Since the loss function is purely based on closeness to the Monge-Ampère solution rather than specific quality measures of the meshes, it cannot totally prevent mesh tangling, as mentioned in the paper.
- The evaluation is done on small meshes. It is unclear how well the method performs on larger meshes (say more than 10K vertices).

---

> ### Author Response · Authors · 2022-08-02
> **Response to Reviewer 2ifw**
>
> Thank you for your detailed and helpful comments and suggestions.
>
> - **Non-planar domains:** We would like to clarify that, the Monge-Ampère method is not specifically designed for non-planar domains. In this paper, we focus on the more fundamental and more widely applied planar domain problems. We apologize for the misleading cited literature and have removed them in the updated manuscript.
> - **Hardware and software setting details:** The experiments were run on an 8-core CPU. The implementation of the Monge-Ampère method naturally supports parallelization since [Firedrake](https://www.firedrakeproject.org/) itself supports MPI (Message Passing Interface), which uses PETSc and libblas-3.7.1 as the underlying linear algebra library. However, we only used single processing across the experiments, because we found that for our experiments, multiprocessing actually did not provide a performance boost. We would like to thank you for proposing these questions, and have updated the manuscript to provide more detailed hardware and software settings for clarification. We would also like to highlight that, the novelty and contribution of this work lie more at the algorithm level.
> - **How to evaluate mesh quality:** Thank you for proposing this very interesting and important topic.
>     - The golden standard to evaluate the quality of a mesh is the corresponding numerical error to solve the PDE. It is indeed possible to take the numerical error as the loss function to train a neural mesh deformer, by implementing the adjoint method, so that the loss can backpropagate through the PDE solver into the neural network. However, it is practically impossible, because in this way, we need to solve a PDE problem for each sample during training.
>     - There are also some geometry-based metrics to evaluate the quality of a mesh. They are very efficient to compute, but don't always agree with the numerical error target. A typical example is that, the aspect ratio metric is frequently used to evaluate the quality of a mesh, but an anisotropic mesh can give better numerical performances for specific PDE problems.
>     - Therefore, taking high-quality adapted meshes generated by advanced mesh adaptation methods to perform supervised learning seems to be a good solution. In our case, we chose the Monge-Ampère method. But we would like to emphasize that, there should be no problem to replace the Monge-Ampère method with other advanced r-adaptation methods.
> - **Mesh tangling avoidance:** We would like to clarify that, mesh tangling is not alleviated by the loss, but by our specifically designed network structure. Even if the loss function is not well optimized, the mesh deformation provided by the neural network is still highly unlikely to cause mesh tangling.
>     - Intuitively speaking, the Neural-Spline based model gives a diffeomorphism so that no two points in the original domain will be mapped to the same location. The GAT based model natively guarantees that all vertices can only move within the convex hull of their neighbors.
>     - Besides alleviating mesh tangling, the proposed models also hold the capability of keeping boundary consistency and generalization to mesh with different resolutions. Note that these requirements are vital for mesh movement, but cannot be trivially achieved by straightforward neural network structures.
> - **Experiment scale:** we totally agree with you that the proposed methods need to work on large-scale problems to be really practical. On the other hand, we would also like to highlight that, this is to the best of our knowledge the first trial towards a learning-based end-to-end mesh movement method, and our experiments can serve as proof of concept to demonstrate its effectiveness.

---

### Official Review · Reviewer_9Nru · 2022-07-19

**Rating:** 6
**Confidence:** 2
**Soundness:** 3 good
**Presentation:** 3 good
**Contribution:** 3 good

**Summary:**

This paper proposes a new technique for moving the vertices of a mesh that is used as the underlying discretization for solving a PDE in order to better capture relevant details without increasing complexity. This is achieved by designing a neural spline-based, and also a graph attention network-based model to output the optimized node positions. Direct supervision coming from the Monge-Ampère method is utilized.

**Questions:**

- As far as I understand, with the GAT-based variant, it is not possible to guarantee non-self-intersecting meshes/mesh tangling. I was expecting an analysis of the behaviour of the generated maps for this characteristic, but there is none.

- I was a bit disappointed by the irregular domain experiments, which is the main motivation for using GAT. Heptagon is not that irregular. I suggest including more examples with ideally real-world-like domains.

**Limitations:**

Please see above.

**Strengths And Weaknesses:**

Strenghths:

- To my knowledge, this is the first method that proposes to optimize for node positions with direct supervision in the context of solving PDEs.
- Can potentially inspire a line of research for re-meshing.

Weaknesses:

- Although the general technique is well-motivated, it is not clear why the particular choices of network architectures make sense. Most of the methods section is about listing the various design choices without justifications.

---

> ### Author Response · Authors · 2022-08-02
> **Response to Reviewer 9Nru**
>
> Thank you for your detailed and helpful comments and suggestions.
> - **Network architecture choice:** our specific network design is out of the requirements of the mesh movement task, which are: alleviating mesh tangling, boundary consistency, and generalization to mesh with different resolutions. Note that these requirements are vital for mesh movement, but cannot be trivially achieved. In the experiments, we provide comparisons against baseline approaches to show the effectiveness of our design.
>
>
> - **Mesh tangling guarantee for M2N-GAT:** Your understanding is accurate that introducing the GAT mechanism into the model can not theoretically avoid mesh tangling. That is why in the paper we only claim the proposed network structure can "alleviate mesh tangling", and we have listed "theoretical analysis on mesh tangling avoidance" as one of the future directions in the paper. On the other hand, we should note that, by keeping each vertex within the convex hull of its neighbors, mesh tangling can be greatly alleviated, which is empirically proven by the experiments.
>
>
> - **Irregular domain experiment:** We didn't perform experiments on highly-irregular domains, because frankly speaking, currently there are no traditional r-adaptation methods that can work satisfyingly on highly-irregularly domains to provide high-quality supervised signals. The theory behind the Monge-Ampère method requires the assumption of a convex domain, hence the performance of the Monge-Ampère method degrades heavily for highly-irregular domains. Other r-adaptation methods such as the spring model based method still encounter mesh tangling problems. On the other hand, we would like to highlight that, regular domain problems or irregular but still convex domain problems can already cover a wide range of scenarios.

---

### Author Response · Authors · 2022-08-02
**Revised Manuscript**

We would like to sincerely thank all the reviewers for your professional and constructive comments. We have replied to your questions and comments in detail respectively. We have also revised the manuscript accordingly to address the proposed issues, including the experimental settings, cited literature, etc.

---

### Meta-Review · Area_Chair_88T3 · 2022-08-24

**Recommendation:** Accept
**Confidence:** Less certain

**Metareview:**

Reviews for this paper were somewhat mixed, but overall the AC agrees with reviewer h6GN and others that the idea in this work is creative and could inspire future work.  In particular, while the experiments do not display diversity in terms of mesh density/shape, the overall approach is compelling and fairly well verified by the experiments.  It seems reasonable for NeurIPS to take a risk on publishing this work in hopes that it will be valuable for future PDE solution machinery that can address some of the missing finer points.

In the revised camera-ready version of the paper, please acknowledge the shortcomings/limitations of the work (e.g., use of Monge-Ampere in contrast to other methods and use of simple 2D problems) and say something about how you might address these limitations.

**Award:**

No

---

### Decision · Program_Chairs · 2022-09-14

Accept